# Fine-scale movement patterns and habitat selection of little owls (*Athene noctua*) from two declining populations

**Martin Mayer**[1]*, **Martin Šálek**[2,3], **Anthony David Fox**[1], **Frej Juhl Lindhøj**[1], **Lars Bo Jacobsen**[4], **Peter Sunde**[1]

**1** Department of Bioscience, Aarhus University, Aarhus, Denmark, **2** Czech Academy of Sciences, Institute of Vertebrate Biology, Brno, Czech Republic, **3** Faculty of Environmental Sciences, Czech University of Life Sciences Prague, Prague, Czech Republic, **4** Center for Macroecology, Evolution and Climate, Globe Institute, University of Copenhagen, Copenhagen, Denmark

* martin.mayer@bios.au.dk

**Data Availability Statement:** Relevant data is available at DOI: 10.5061/dryad.k3j9kd57m.

**Funding:** This work was supported by the Environmental Protection Agency, Ministry of

## Abstract

Advances in bio-logging technology for wildlife monitoring have expanded our ability to study space use and behavior of many animal species at increasingly detailed scales. However, such data can be challenging to analyze due to autocorrelation of GPS positions. As a case study, we investigated spatiotemporal movements and habitat selection in the little owl (*Athene noctua*), a bird species that is declining in central Europe and verges on extinction in Denmark. We equipped 6 Danish food-supplemented little owls and 6 non-supplemented owls in the Czech Republic with high-resolution GPS loggers that recorded one position per minute. Nightly space use, measured as 95% kernel density estimates, of Danish male owls were on average 62 ha (± 64 SD, larger than any found in previous studies) compared to 2 ha (± 1) in females, and to 3 ± 1 ha (males) versus 3 ± 5 ha (females) in the Czech Republic. Foraging Danish male owls moved on average 4-fold further from their nest and at almost double the distance per hour than Czech males. To create availability data for the habitat selection analysis, we accounted for high spatiotemporal autocorrelation of the GPS data by simulating correlated random walks with the same autocorrelation structure as the actual little owl movement trajectories. We found that habitat selection was similar between Danish and Czech owls, with individuals selecting for short vegetation and areas with high structural diversity. Our limited sample size did not allow us to infer patterns on a population level, but nevertheless demonstrates how high-resolution GPS data can help to identify critical habitat requirements to better formulate conservation actions on a local scale.

## Introduction

Land use change and habitat loss, largely driven by agricultural expansion and intensification, are a major cause of global animal population declines and biodiversity loss [1, 2]. Given these widespread, dramatic declines in wildlife populations, identifying key species habitat requirements is crucial to formulate effective conservation actions. This can be achieved on different

Environment and Food Protection of Denmark (project name: "Udredning af gunstig naturtilstand for kirkeugle"), Ministry of the Environment of the Czech Republic (project no. 200129), research aim of the Czech Academy of Sciences (RVO 68081766), and by the program of the Strategy AV 21. The funders had no role in study design, data collection and analysis, decision to publish, or preparation of the manuscript.

**Competing interests:** The authors have declared that no competing interests exist.

spatial scales, ranging from species distribution modelling on a landscape scale to investigating the selection of specific resources within an individuals' home range [3–5]. Recent decades have seen a surge in technological advances allowing the collection of detailed information on movements of individuals, based on high-frequency relocations, which are crucial to study the mechanistic aspects of individual habitat selection and provide insights to make effective conservation management decisions [6, 7].

Although high-frequency relocations provide increasingly detailed data, they are often highly spatiotemporally autocorrelated [8], which can lead to analytical problems, as dependency between relocations produces more similar values than expected by chance, underestimating the true variance [6]. This problem is especially important in third-order habitat selection, when comparing random (available) positions that lack an autocorrelation structure with observed GPS positions (that are autocorrelated), because it can lead to unreliable results [9]. Previously, independence between relocations was often ensured by subsampling data to reduce autocorrelation, by using a large time lag between successive relocations, or by post hoc variance inflation to adjust standard errors after parameter estimation [6, 10, 11]. However, this can lead to the loss of biological information. A newer approach is to simulate a random movement trajectory with the same autocorrelation structure as the observed animal movement trajectory, using correlated random walks [12].

Here, we conducted a case study on the little owl (*Athene noctua*) to investigate how high-frequency GPS data can be used to investigate spatio-temporal movement patterns and habitat selection to inform conservation practices. Little owls are nocturnal generalist predators, feeding on small mammals, birds, amphibians and a wide range of invertebrates, depending on prey availability [13–17]. Prey are captured from perches or by walking on the ground, and bare soil or short (<10 cm) grass vegetation is preferred for foraging [18, 19]. Selection for different prey items depends on local weather conditions, with owls hunting small mammals in dry weather and earthworms during wet weather [20].

Little owls are widely distributed across large parts of Europe, Asia and North Africa and are categorized as 'least concern' by the International Union for Conservation of Nature [21]. However, local populations are declining in many European countries, with agricultural intensification suggested as the main cause, reducing both nest site and food availability [22–25]. In Denmark, the species has declined from being locally the most abundant owl species in the 1970s to less than 100 pairs in the 2000s and 11 breeding pairs in 2020 [26]. This decline was linked to reduced reproductive success caused by food limitation during the breeding season, likely the result of agricultural intensification [27–29]. Since 2009, most remaining Danish little owl pairs have been food-supplemented in the breeding season (with food supplementation constituting about 38% of the caloric intake of little owl chicks: [30]), that has led to improved chick survival, fledgling production [28, 31], and reduced parental provisioning effort [32]. Feeding programmes may raise breeding success sufficiently to reverse local population declines in the short-term [33], but to achieve naturally sustainable populations, reestablishment of adequate habitat quality and prey availability are needed at relevant spatial scales. Information regarding space use is therefore crucial as a basis for planning habitat restoration initiatives, i.e., to identify essential profitable habitats over which owls forage during the breeding season [34, 35].

We compared nightly little owl space use, spatio-temporal movement patterns, and habitat selection in surviving territories in Denmark with a local population in the Czech Republic that is also declining [23], but not food-supplemented. We chose this comparison to contrast space use patterns among individuals of these two populations, both of which were located in intensively cultivated farmland, but differed in the composition of crop types and regarding supplementary feeding, with the aim to understand space use to inform habitat restoration

activities. Home range sizes (as measure of habitat quality [36]) and displacement distances of Danish little owls have previously been shown to be at least twice the size of other populations [25, 37]. Thus, we predicted that Danish owls would forage over considerably larger areas than Czech owls. Moreover, little owls (like other raptor species) show sex related differences in parental effort, with males the more important food provider, while females stay closer to the nest [27]. Consequently, we predicted that males explore larger areas, move longer distances than females and respond more strongly to habitat quality than females. Finally, we predicted that little owls adjust spatial movement patterns in response to weather conditions and the time of the night, because these factors might affect prey availability. Regarding habitat selection, we predicted that owls generally select for areas that are associated with high prey densities and allow for successful foraging, i.e. comparatively short vegetation of pastures.

## Material and methods

### Study areas and data collection

The Danish study area in Østhimmerland, northern Jutland (56.868˚ Lat, 9.161˚ Lon; Fig 1) was dominated by agricultural land, >80% of the surface being cropland or pastures. At the time of the study (2019), the Danish little owl population was estimated at ca. 11 breeding pairs [26]. Five owl pairs bred within a 2x2 km area (shortest distance between neighboring pairs: 400 m), comprising the last cluster of contiguous little owl pairs in Denmark. Owls bred in nest boxes at or within buildings and were food-supplemented with ca. 5 newly hatched poultry chicks per day. The Czech study areas were located in northern Bohemia (50.460˚ Lat, 13.560˚ Lon and 50.287˚ Lat, 14.154˚ Lon; Fig 1), the current core area of the little owl distribution in the Czech Republic [38, 39]. Czech little owls were not food-supplemented. The landscape structure and land-use composition has also been highly modified by agricultural intensification [40], causing large-scale reduction of natural and semi-natural non-cropped habitats, as well as an increase in soil fertilization and pesticide use [41]. Average local little owl population density in the study area in 2020 was 0.41 calling males/10 km$^2$ (Šálek, unpublished data). While Danish owls persistently fail to produce $\geq 2$ fledglings without food supplementation [28], Czech owls experienced their most successful breeding season on record in the study year, 2020 [42; Šálek, unpublished results].

We captured 6 adult owls from both the Danish (June 2019) and Czech populations (June 2020; Table 1) using mist-nets or traps baited with dead chickens (Denmark) or live mice (Czech Republic). This period coincided with the fledgling period in both areas. Captured owls were weighed, measured, sexed (based on brood patch and body mass) and fitted with a GPS backpack [20]. We attached Gypsy 5 GPS loggers (Technosmart Europe srl., Rome; total mass including Teflon backpack harness of ca. 3.2 g; constituting <2.5% of the owls' body mass) which provided positional accuracy down to 2 ± 4 m. GPS units were programmed to record one position per minute, and to start recording positions the night after capture (avoiding potential capture effects). The tags of the first four owls recorded positions between 20:00 and 05:00 local time (Central European Summer Time, CEST; i.e., UTC + 2). As this setting resulted in fewer positions and survey nights than expected (likely because the GPS was activated while the birds were still inside buildings, resulting in excessive power consumption), for the remaining birds, the logging interval was narrowed to between 21:00 and 04:00 local time. After seven to nine days, owls were recaptured, tags retrieved and data downloaded. We obtained hourly weather data (precipitation categorized as yes/no and mean temperature) from Rebild municipality for the Danish study area (https://www.dmi.dk/vejrarkiv/) and from Václav Havel Airport Prague and Ústí nad Labem weather stations for the Czech study areas (https://www.timeanddate.com/weather).

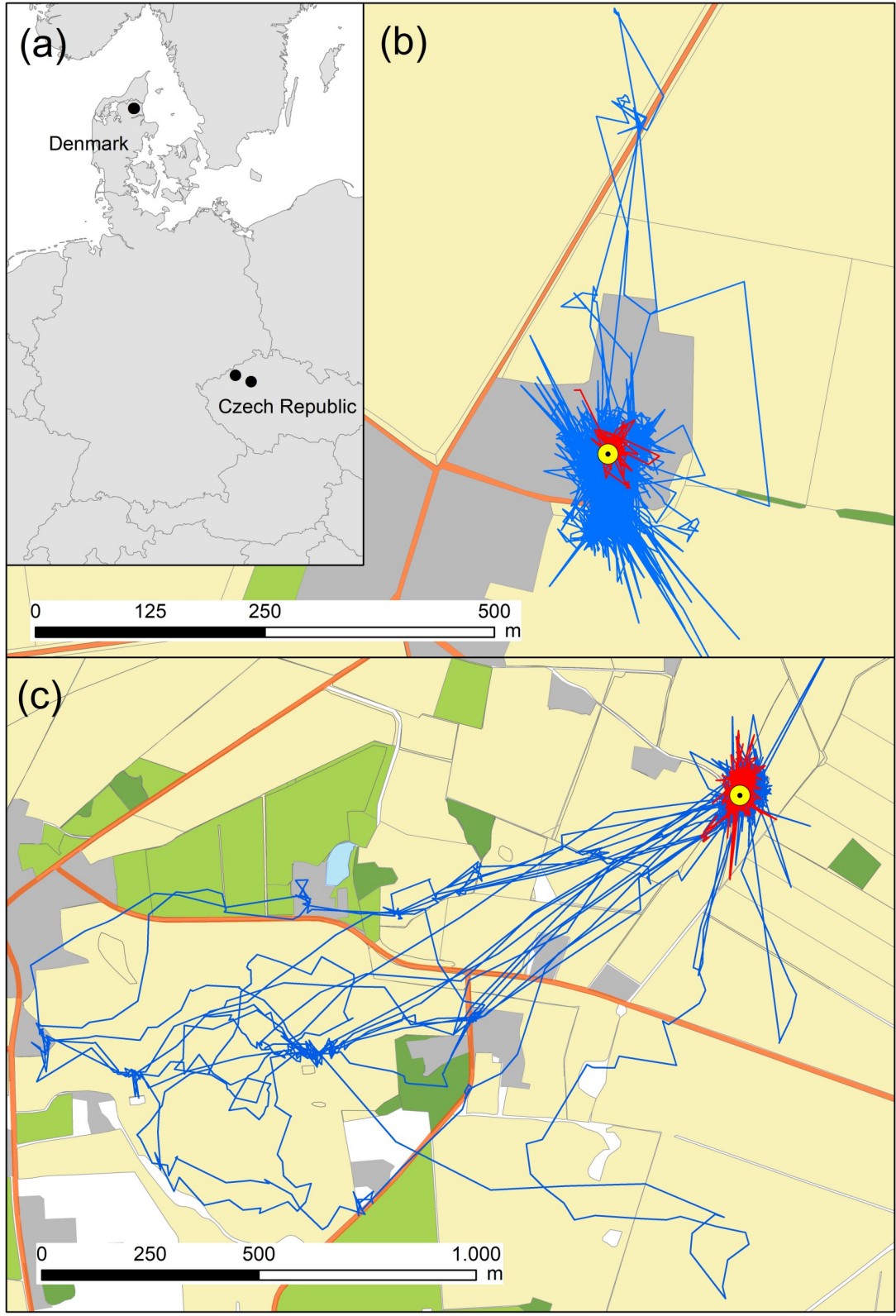

**Fig 1. Overview of the study areas and movement trajectories of little owls.** (a) Locations of the study areas (black dots) in Himmerland, Denmark and northern Bohemia, Czech Republic. Main maps show exemplary movement trajectories of females (red lines) and males (blue lines) for individuals in the Czech Republic (b) and Denmark (c). The nest location is shown as a

yellow dot. Land cover classes: built up = grey; forest = dark green; pasture = light green; arable = beige; roads = orange. The figure was created using ArcMap 10.1 (Esri, Redlands, CA, USA, http://www.esri.com/arcgis/about-arcgis), using open access maps (http://www.naturalearthdata.com/).

### Ethics declaration

All capture and handling procedures were approved by the Copenhagen Bird Ringing Centre and the Natural History Museum of Denmark for the Danish study, and the Bird Ringing Centre, National Museum, Czech Republic for the Czech study. Our study met the ASAB/ABS Guidelines for the treatment of animals in behavioral research and teaching [43]. All methods were performed in accordance with the relevant guidelines and regulations. No owls were injured during capture and handling, nor did the tags or the harness inflict any observable lesions, bruises or damage to skin or plumage.

### Data preparation and statistical analyses

**Spatial behavior.** GPS positions obtained with <4 satellites and a horizontal dilution of precision (HDOP) value >5 were removed from the data to remove GPS locations with a large location error [44]. In general, GPS units only recorded data from active owls, i.e. individuals that were outside the nest box/buildings (because the GPS could not acquire fixes inside buildings).

We obtained the following variables from the raw GPS data: (1) *nightly space use*, calculated as 95% isopleths of Kernel Density Estimates (KDE) separately for each individual and night using the R package 'adehabitatHR' [45], and using the reference method ($h_{ref}$) as smoothing parameter. We only included nights with ≥100 GPS positions and where we had obtained GPS positions for the entire night (i.e., we removed the final night of GPS recordings, if GPS fix acquisition did not last the entire night, because the battery ran out of power). (2) *Distance from nest*, the straight line distance of each GPS position from the nest. (3) *Foraging trips >200 m from the nest*. We calculated the frequency and duration of foraging trips where individuals moved >200 m from their nest. The duration of an individual trip was estimated from the initial nest location until the owl was back at the nest, in both cases defined as a position ≤25 m from the nest. (4) *Hourly displacement*, the sum of all straight line distances between GPS-positions within each hour of the night, and (5) *Nest visitation rate*. Nest visitation rate could only

**Table 1. Overview of GPS-tagged little owls.**

| ID | Territory | Sex | Number of chicks | GPS positions | GPS nights | Mean ± SD nightly 95% KDE (ha) | Mean ± SD displacement per hour (m) | Mean ± SD distance from the nest (m) |
|---|---|---|---|---|---|---|---|---|
| DK1_F | GV16 | F | 3 | 103 | 1 | 1.7 | 912 ± 69 | 45 ± 34 |
| DK1_M | GV16 | M | 3 | 1,029 | 3 | 7.9 ± 1.2 | 1,519 ± 705 | 88 ± 90 |
| DK2_M | GV17 | M | 4 | 2,239 | 6 | 53.2 ± 22.4 | 2,049 ± 855 | 224 ± 277 |
| DK3_M | GV19 | M | 3 | 1,257 | 4 | 30.3 ± 26.5 | 1,424 ± 533 | 167 ± 243 |
| DK4_F | HV5 | F | 4 | 560 | 3 | 1.2 ± 0.3 | 1,093 ± 475 | 33 ± 21 |
| DK4_M | HV5 | M | 4 | 1,903 | 5 | 127.4 ± 75.2 | 1,947 ± 870 | 414 ± 548 |
| CZ1_F | Bres | F | 4 | 980 | 4 | 0.4 ± 0.2 | 522 ± 349 | 16 ± 13 |
| CZ1_M | Bres | M | 4 | 2,782 | 8 | 4.7 ± 2.9 | 1,131 ± 741 | 66 ± 97 |
| CZ2_F | Krov | F | 3 | 279 | 1 | 0.2 | 388 ± 200 | 14 ± 12 |
| CZ2_M | Krov | M | 3 | 2,357 | 7 | 2 ± 1.3 | 929 ± 569 | 49 ± 49 |
| CZ3_F | Mal | F | 5 | 633 | 2 | 9.2 ± 12.5 | 990 ± 497 | 86 ± 158 |
| CZ3_M | Mal | M | 5 | 517 | 3 | 2.4 ± 1.2 | 723 ± 603 | 47 ± 48 |

Summary of telemetry data from little owls GPS-tagged in Denmark (in 2019) and in the Czech Republic (in 2020).

be estimated for males as females stayed too close to the nest to generate reliable estimates. We defined a (presumed) nest visit as one or more GPS-positions ≤25 m from the nest following a location >50 m from the nest, summing number of hourly nest visits separately for each individual and night.

Our sample size was too low for a statistically meaningful comparison of nightly space use. Thus, we simply report the mean ± standard deviation (SD) nightly space use between males and females and between Denmark and the Czech Republic. To analyze the distance from the nest (raw GPS data; log-transformed to meet the assumption of residual normality), we used linear mixed models (LMMs). We included area, sex, time of night (as factor), the number of chicks, mean hourly temperature, precipitation, and the two-way interaction of sex × time of night as fixed effects and owl ID as random intercept (we initially nested owl ID within the territory, but as this did not change the results, we only used owl ID as random intercept). To avoid higher order interactions (i.e. area × sex × time of night), we ran separate analyses for the two countries including, sex, time of night, the number of chicks, precipitation and the interaction of sex × time of night as fixed effects and owl ID as random intercept (S1 Table). For the Czech data, we additionally included the mean hourly temperature, because it was not highly correlated with the time of night (Pearson R = -0.46, p < 0.001). For the Danish data, time of night and hourly temperature were highly correlated (Pearson R = -0.71, p < 0.001), and thus, temperature was not included in this analysis (S1 Table). We analyzed the number of foraging trips per day and trip duration (of trips >200 m from the nest) using LMMs, and included sex, area, and their interaction as fixed effects (S1 Table). To analyze the hourly displacement (response variable, 321 owl hours) we also used LMMs. We included the number of hourly GPS locations to account for biases in the estimation of hourly distances. Additionally, we included the same variables as for the distance from the nest analysis (S1 Table). To analyze hourly nest visits by males (response variable, 258 owl hours), we used a generalized LMM with a log link and a Poisson distribution. We included the number of hourly GPS locations (to account for biases in the estimation of nest visitation rate) and the area as fixed effects and owl ID as random intercept. Finally, we analyzed the proportion of GPS positions within 25 m from the nest (raw data from males only; response variable) using a GLMM with a logit link, including the area as fixed effect and owl ID as random intercept. The proportion of GPS positions within 25 m from the nest might have been underestimated due to reduced GPS fix success rates at the nest, but we assumed this error to be constant across individuals. No fixed effects were correlated (Pearson's r < 0.6 in all cases).

**Habitat use and selection.**   To describe the available habitat, we obtained land cover (vector) data from the Danish Ministry for Food, Agriculture and Fisheries (https://kortdata.fvm.dk/download/Index?page=Markblokke_Marker; accessed 4 April 2020), which included information regarding vegetation type separately for each agricultural field. Further, we monitored each little owl territory in spring 2019 (ca. 2 months before the GPS study) in Denmark and in July 2020 in the Czech Republic (one month after the GPS study), recording the habitat type of each field within a radius of 500 m around each nest. We categorized the land cover data into (1) built up areas (buildings and associated gardens), (2) cereal, (3) fallow areas (vegetated but not used for agricultural production), (4) forest (including tree rows), (5) maize, (6) other arable fields consisting of vegetables, peas, poppy, sugar beet, and alfalfa (merged because they contributed very little area), (7) pastures and grazed/mown surfaces (including football fields), (8) rape and (9) road verges. To describe general habitat composition, we created buffers around the nest location of each territory in 6 distance classes (0–100 m, 100–200 m, 200–300 m 300–400 m, 400–500 m, 500–2000 m) and calculated the proportion of each land cover type within these buffers. Moreover, we investigated differences between the two populations by calculating the average field size within a buffer of 500 m around each nest. We also calculated

Simpson's Index of Diversity as measure of habitat diversity: D = 1 − [Σn(n-1)/ N(N-1)] [46], where in this context n is the area comprised by each land cover type and N is the total area, i.e. a 500 m buffer around each nest location.

We aimed to quantify third-order habitat selection of little owls, i.e. selection of foraging habitat patches [47]. Due to the high GPS fix rate (one position per minute), the GPS data were highly spatially autocorrelated. Thus, we used correlated random walks to create availability data with the same spatial autocorrelation structure as the actual little owl positions using the R package 'SiMRiv' [12]. We simulated correlated random walks by little owls within the maximum distance they had moved from their nest, which served as the central place (S1 Fig). To account for the fact that movements further from the nest are more energetically costly resulting in fewer observed owl positions, we added movement restrictions with increasing distance from the nest [12]. We parameterized the simulation models from real owl trajectory data by numerically finding combinations of input parameters that minimize the differences between the real trajectory and simulated trajectories [12; for details see https:// cran.r-project.org/web/packages/SiMRiv/vignettes/SiMRiv.pdf]. Step length and turning angle distributions were specified from the observed distributions from each individual. Random walks consisted of 10 times the number of observed locations for each individual. We then compared the observed positions to the random points derived from the correlated random walks [48]. The use of step selection functions was not applicable in this case, because average step lengths were too short to include unused areas that were available to owls [49]. Similarly, creating available positions exclusively within the owls' home ranges (95% KDE) would have excluded areas that were clearly available, but unused.

We assigned all observed (owl) and available (correlated random walk) positions to the habitat categories, and calculated the distance of each position to the closest field edge and the closest road. The distance to field edges included features that were important for hunting little owls (proximity to perches, such as fence lines, trees and hedgerows). We fitted logistic resource selection functions [50], using generalized linear mixed-effects models with a binomial distribution and a logit link, using the R package 'lme4' [51], separately for Czech and Danish owls and for females and males with the response variable being the observed owl positions (= 1) and available random walk positions (= 0). We included the land cover type and distance to field edges as fixed effects and owl ID as random intercept. For the Danish owls, we additionally included the distance from roads as a fixed effect (Table 2). This was highly correlated with distance from field edges for Czech owls. For males only (because they moved further from the nest), we ran an additional model including the two-way interaction of habitat × nest distance (i.e. in addition to the above-mentioned variables) to investigate, if habitat selection changed with distance from the nest.

### Model selection

Model selection for all analyses was based on a stepwise variable selection using Akaike's information criterion corrected for small sample size ($AIC_c$) [52], selecting the model with the lowest $AIC_c$ [53], using the R package 'MuMIn'[54]. Parameters that included zero within their 95% CI were considered uninformative [55]. We validated the most parsimonious models by plotting the model residuals versus the fitted values [56]. All statistical analyses were carried out in R 3.6.0 [57]. Values are given as mean ± SD if not stated otherwise.

### Results

In total, we obtained 14,639 GPS positions of 6 individuals in Denmark (food-supplemented; 4 males and 2 females) and 6 owls in the Czech Republic (not food-supplemented; 3 males and 3

**Table 2. Analyses of habitat selection by little owls.**

| Variable | Danish females | Danish males | Czech females | Czech males |
|---|---|---|---|---|
| Intercept | **-4.57 (-5.48, -3.66)** | **-4.15 (-4.79, -3.52)** | **-4.58 (-4.95, -4.2)** | **-3.18 (-4.06, -2.3)** |
| Land cover Built up | **2.84 (2.58, 3.09)** | **4.75 (4.64, 4.86)** | **2.64 (2.32, 2.95)** | **2.84 (2.61, 3.07)** |
| Land cover Fallow | | **-1.26 (-2.25, -0.27)** | -0.25 (-0.99, 0.48) | **3.39 (3.12, 3.66)** |
| Land cover Forest | | **2.81 (2.68, 2.93)** | **2.75 (1.22, 4.28)** | **-2.98 (-4.41, -1.56)** |
| Land cover Maize | | **5.46 (5.24, 5.68)** | -13.2 (-96.82, 70.43) | **-1.1 (-1.36, -0.83)** |
| Land cover Other arable | | **1.49 (1.3, 1.69)** | -0.59 (-1.77, 0.59) | **0.59 (0.32, 0.87)** |
| Land cover Pasture | **1.46 (0.9, 2.02)** | **1.11 (1.01, 1.22)** | **2.01 (1.5, 2.51)** | **0.33 (0.04, 0.62)** |
| Land cover Rape | **3.71 (2.23, 5.18)** | -0.23 (-0.48, 0.02) | **3.34 (2.96, 3.73)** | **-2.28 (-2.59, -1.96)** |
| Land cover Road verges | -0.8 (-2.79, 1.18) | **1.84 (1.65, 2.02)** | **1.82 (1.29, 2.34)** | **0.35 (0.06, 0.63)** |
| Distance to field edge | **-0.35 (-0.61, -0.08)** | **-0.41 (-0.46, -0.36)** | **0.2 (0.1, 0.3)** | **-0.07 (-0.13, -0.01)** |
| Distance to road | **0.76 (0.27, 1.25)** | **-0.07 (-0.11, -0.03)** | | |

Beta coefficients and 95% confidence intervals (in brackets) for the analyses of habitat selection by female and male little owls separately for Denmark and the Czech Republic. Cereal was used as reference level, i.e. habitat selection of the other land cover types are presented in relation to cereal. Informative parameters (95% confidence intervals not overlapping zero) are in bold.

females), with GPS units obtaining data for 1–8 nights (Table 1). The number of GPS positions per night ranged between 87 and 531 (311 ± 119). Excluding GPS positions with <4 satellites and HDOP >5 led to the removal of 1,388 positions (9% of the data).

## Spatial behavior

**Nightly space use.** Nightly 95% KDEs varied in size from 1–197 ha (26.1 ± 46.3 ha; median: 4.0 ha) and average nightly space use was larger in Denmark compared to the Czech Republic (41.8 ± 58.6 versus 3.2 ± 3.3 ha). Male nightly space use was an order of magnitude larger than female space use in Denmark (61.8 ± 64.3 versus 2.0 ± 1.0 ha), but there was no difference in the Czech Republic (3.4 ± 1.4 versus 3.3 ± 5.0 ha).

**Distance from nest.** During their activity period, little owls moved between 0 and 1,753 m from the nest (139 ± 277 m; median: 37 m). Owls moved further from the nest in Denmark compared to the Czech Republic (Fig 2, S1 and S2 Tables). This difference was caused by males, which on average forayed 4-fold further from their nest in Denmark compared to the Czech Republic (248 ± 379 versus 58 ± 78 m), whereas female nest distances were comparable in Denmark and the Czech Republic (34 ± 34 versus 38 ± 95 m). In Denmark, females generally stayed close to the nest independent of the time and males moved further from the nest with two peaks around 2300 and 0300 h. (Fig 2, S2 Table). In the Czech Republic, males also moved further from the nest around 2200 and 0430 h, and were generally further from the nest than females (Fig 2). However, one female also made two longer forays further from the nest (Fig 2). Moreover, in the Czech Republic individuals moved further from the nest with precipitation and increasing temperature (S2 Table). In the analysis of Danish owl data, number of chicks and precipitation were not included in the most parsimonious model (S1 Table).

**Foraging trips >200 m from the nest.** We recorded 93 foraging trips that extended >200 m from the nest. Males conducted ca. 3-fold more foraging trips per night than females (2.5 ± 1.7 versus 0.8 ± 0.8 trips), but there was no measurable difference between areas (S1 and S3 Tables). These trips lasted on average 37 ± 32 min (range: 3–132 min). Trip duration was affected by the interaction between area and sex, with male trips shorter than female trips in the Czech Republic (18 ± 12 versus 36 ± 29 min), but longer in Danish males compared to Danish females (49 ± 35 versus 14 ± 13 min; S1 and S3 Tables).

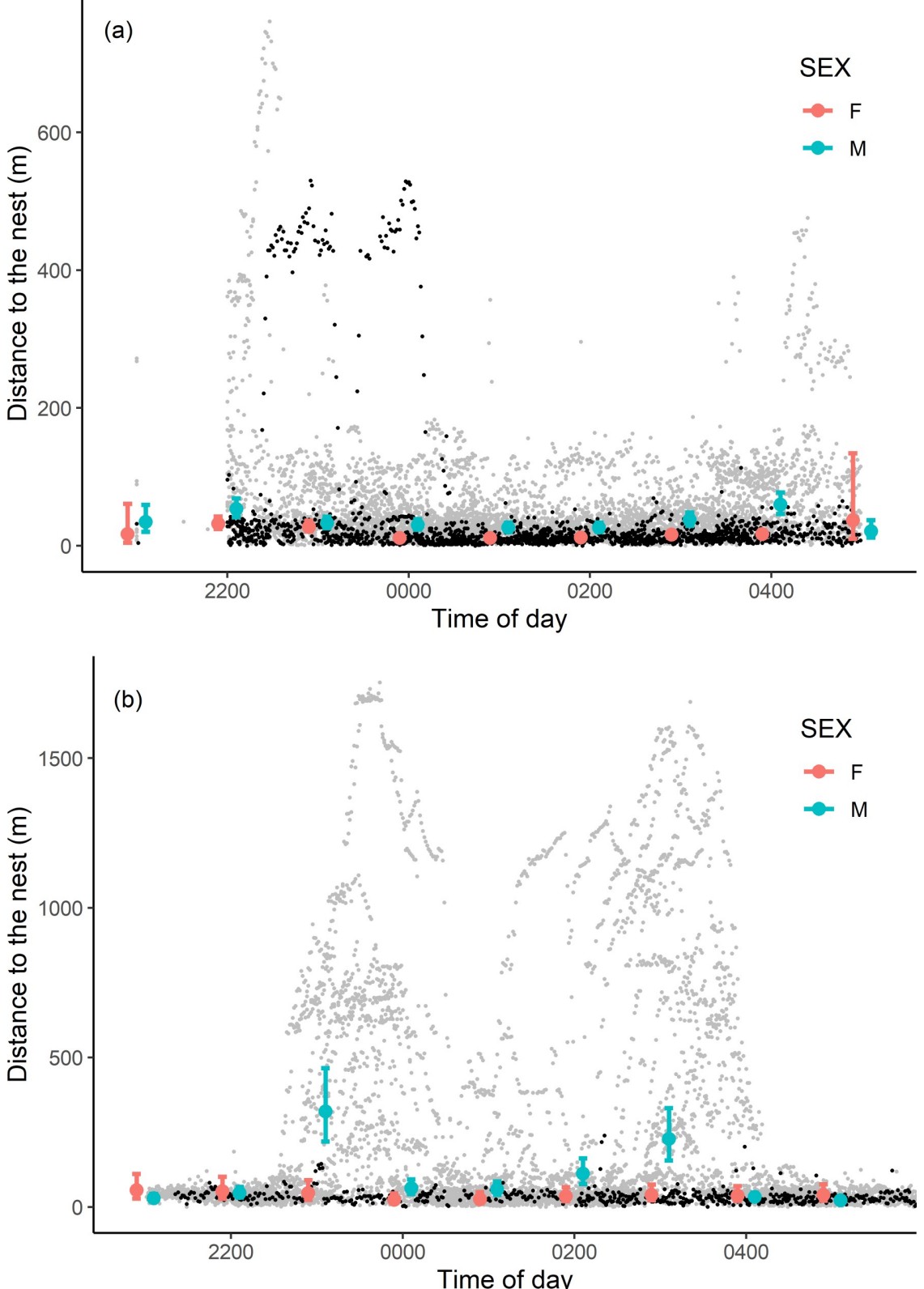

**Fig 2. Distance to the nest in relation to the local time of day (CEST) shown separately for little owls tracked in the Czech Republic (a) and Denmark (b).** Model estimates (large dots) and 95% confidence intervals (bars) are shown separately for females

(F; orange) and males (M; cyan). Raw data are also shown separately for females (black dots) and males (grey dots). Note the different scales of the y-axes between the Czech Republic and Denmark.

**Hourly displacement.** We obtained on average 41 ± 19 GPS positions per hour. The hourly displacement ranged from 2 to 4,797 m (1,232 ± 819 m), and increased with increasing number of GPS positions (obtained per hourly interval) and the number of chicks (S1 and S2 Tables). Danish owls moved almost twice as far per hour compared to owls from the Czech Republic (females: 1,057 ± 428 versus 620 ± 429 m; males: 1,796 ± 812 versus 993 ± 670 m), and displacement distances depended on the time of night, with individuals moving more around 2100 and 0100 h (Fig 3, S2 Table). Temperature, precipitation, sex and the interaction of area and sex were retained in the most parsimonious model in all analyses, but were mostly uninformative (S2 Table).

**Nest visitation rate.** Male owls visited the nest on average 13 times per night in both populations, and the number of hourly nest visits did not differ between Denmark and the Czech Republic (1.7 ± 1.6 versus 2.0 ± 1.5, Estimate ± SE: -0.11 ± 0.15, 95% CI: -0.47; 0.22). However, Czech males spent more time within 25 m from the nest compared to Danish males (0.34 ± 0.47 versus 0.18 ± 0.39, Estimate ± SE: 0.95 ± 0.28, 95% CI: 0.40; 1.49).

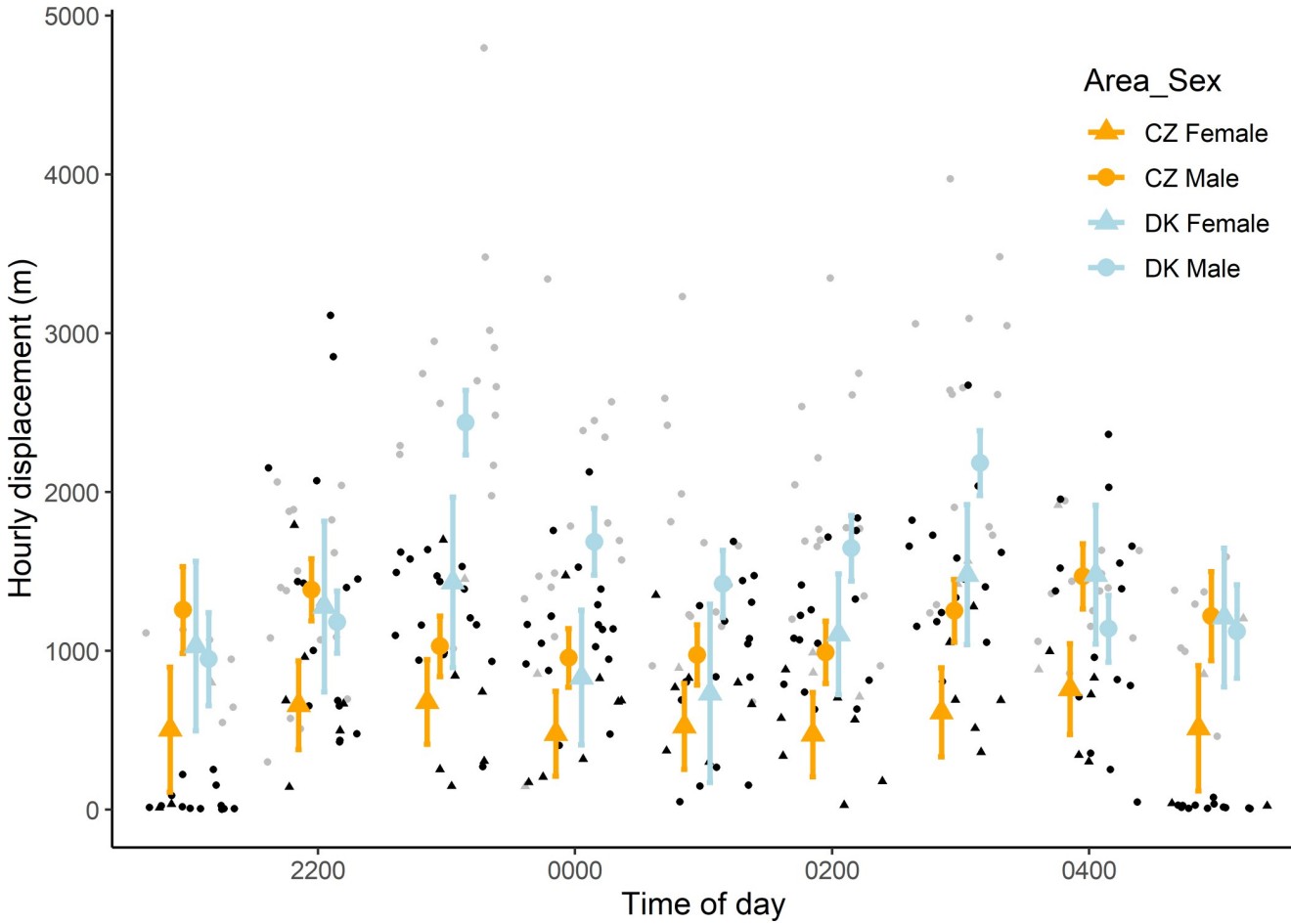

**Fig 3. The predicted displacement per hour (large symbols) and 95% confidence intervals (bars) separately for Czech (orange) and Danish (light blue) little owls.** Raw data are also shown separately for Czech (black) and Danish (grey) owls. Females are shown as triangles and males as circles.

## Habitat use and selection

Mean field sizes were comparable in the Czech Republic and Denmark (4.8 ± 12.6 ha versus 4.9 ± 5.9 ha; unpaired t-test: t = -0.02, df = 162, p = 0.98). Similarly there was no statistical difference regarding habitat diversity between Czech and Danish owl territories (Simpson's Index of Diversity: 0.68 ± 0.03 versus 0.62 ± 0.11, Mann–Whitney test: W = 8, p = 0.63). Available land cover types based on buffers around the nest location changed with distance from the nest, with the Czech area dominated by maize, cereals and built up areas, and in Denmark by cereal, pastures and rape (Fig 4). Habitat use also changed with distance from the nest, with

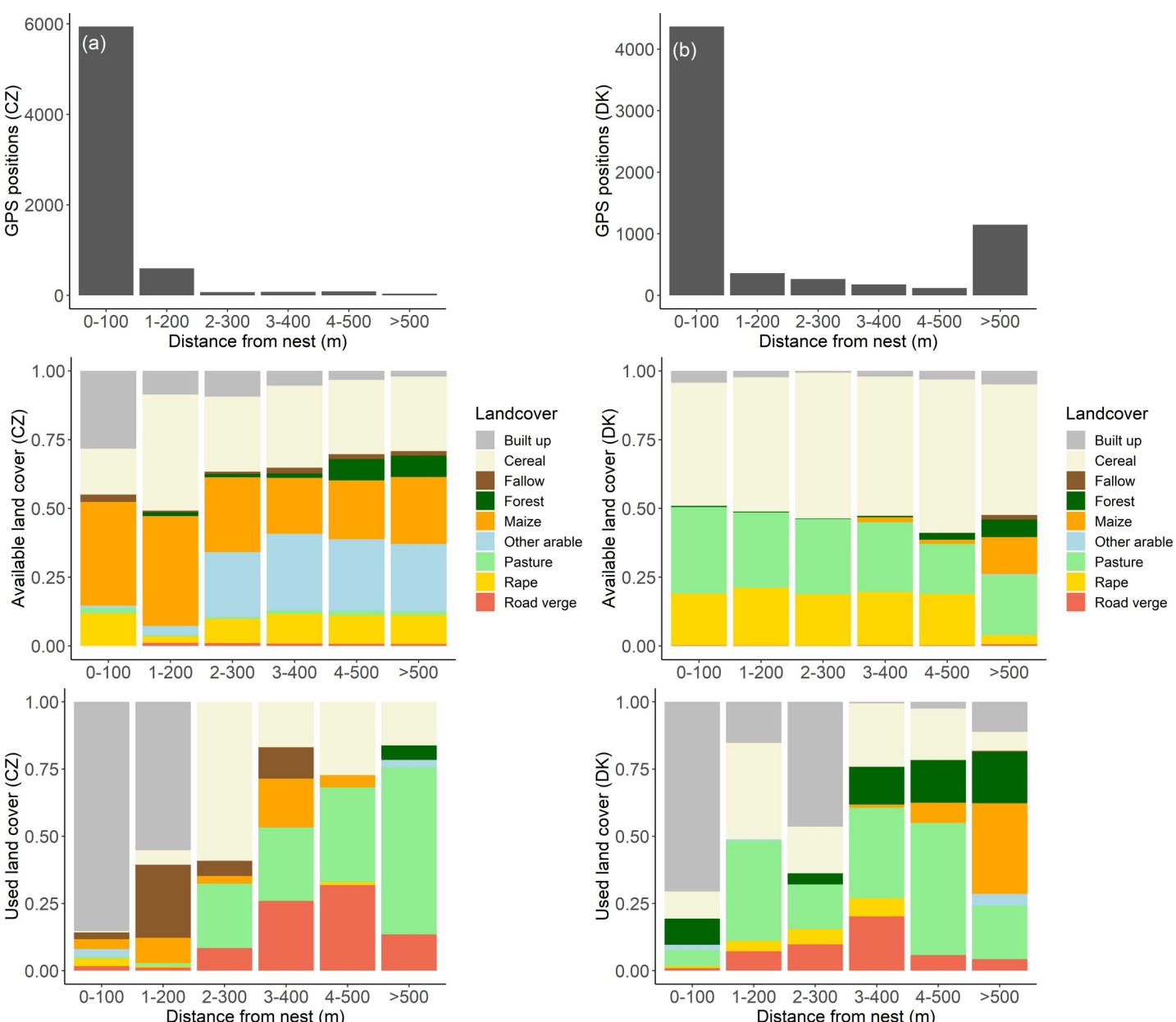

**Fig 4. The number of GPS positions (top), available (middle) and used (bottom) land cover types shown at different distance intervals from little owl nests separately for the Czech Republic (a; left panel) and Denmark (b; right panel).** Data were pooled across individuals for each area. Note that the available habitat was calculated from buffer zones around each territories' nest location (i.e., from land cover vector layers and not from correlated random walks). Moreover, the last distance category in Denmark is 500–2000 m from the nest compared to 500–1000 m for Czech owls, because Danish owls moved further from the nest.

built up areas being mostly used in close nest proximity, whereas cereal, pastures and road verges were used with increasing distance from the nest (Fig 4). In general, built up areas were by far the most used land cover type, making up 79% of all GPS positions in the Czech Republic, followed by fallow (5%), maize (4%), rape (3%), road verges (2%), pasture (2%), cereal (2%), and other land cover (3%). In Denmark, built up areas were also the most used land cover type (53%), followed by pasture (12%), cereal (12%), forest (11%), maize (6%), road verges (3%) and other land cover (3%).

Regarding habitat selection (comparing used positions to available positions obtained from correlated random walks; S1 Fig), all owls (of both sexes and study areas) strongly selected for built up areas, avoided cereal, and showed no selection or avoidance for pastures and road verges. For the other land cover types, habitat selection partly differed between females and males, and between the two study areas. In both areas, rape was avoided by males, but selected by females. Maize and forest were selected by Danish males and avoided by Czech males, and other arable crops were neither selected nor avoided by males of both populations. Fallow, maize, and other arable crops were either avoided by females (in both populations) or not available; i.e., not present in proximity to the nest (Fig 5). Forest was neither selected for nor avoided by Danish females and not available to Czech females. Danish males selected for proximity to roads whereas Danish females avoided proximity to roads (this variable was not included in the analyses of Czech owls), and all owls, apart from Czech females, selected for proximity to field edges (Table 2). Finally, male owls in Denmark increasingly selected for maize with increasing distance from the nest at the expense of the other land cover categories, whereas there was no clear pattern for males in the Czech Republic (S2 Fig).

## Discussion

Our study shows that high-resolution GPS data can provide a detailed understanding of fine-scale spatio-temporal movement patterns of little owls. For example, the majority (80%) of foraging trips >200 m from the nest lasted <1 h and 30% of these trips lasted <15 min. Thus, fix rates of >15 min between successive GPS positions might lead to substantial loss of detail regarding spatio-temporal movement patterns [58]. However, there are obvious trade-offs between battery life and the spatial resolution of GPS data, as GPS units in our study only recorded between one and eight nights of data. Deploying some study animals with GPS units that record data at a high fix rate (here one position per minute) can be useful to identify biologically meaningful intervals between GPS positions for future studies. In our case, a 10–15 min fix rate would likely increase battery life to cover most of the breeding period while allowing to record sufficiently detailed data. Moreover, our study shows that high-resolution GPS data is useful for habitat selection analyses when observed data is compared to simulated movement trajectories based on correlated random walks to account for spatiotemporal autocorrelation of the data [6, 12].

Given the low number of tagged birds and short duration of data collection, our results have to be taken cautiously and cannot be extrapolated to the population level. Nevertheless, the high spatio-temporal resolution of the locations provide new insight into how little owls move during provisioning of young. Differences in owl movements were predominantly driven by males, which in Denmark had much larger nightly space use areas than females, caused by larger distances from the nest, more foraging trips, and larger distances moved, whereas movement differences between the sexes and pairs in the Czech Republic were not as pronounced. Greater movements by males are in line with previous findings in little owls and other raptors, suggesting that males are the main food providers, whereas females forage in close proximity to the nest, where they care for the young [27, 59]. Movements of male owls in

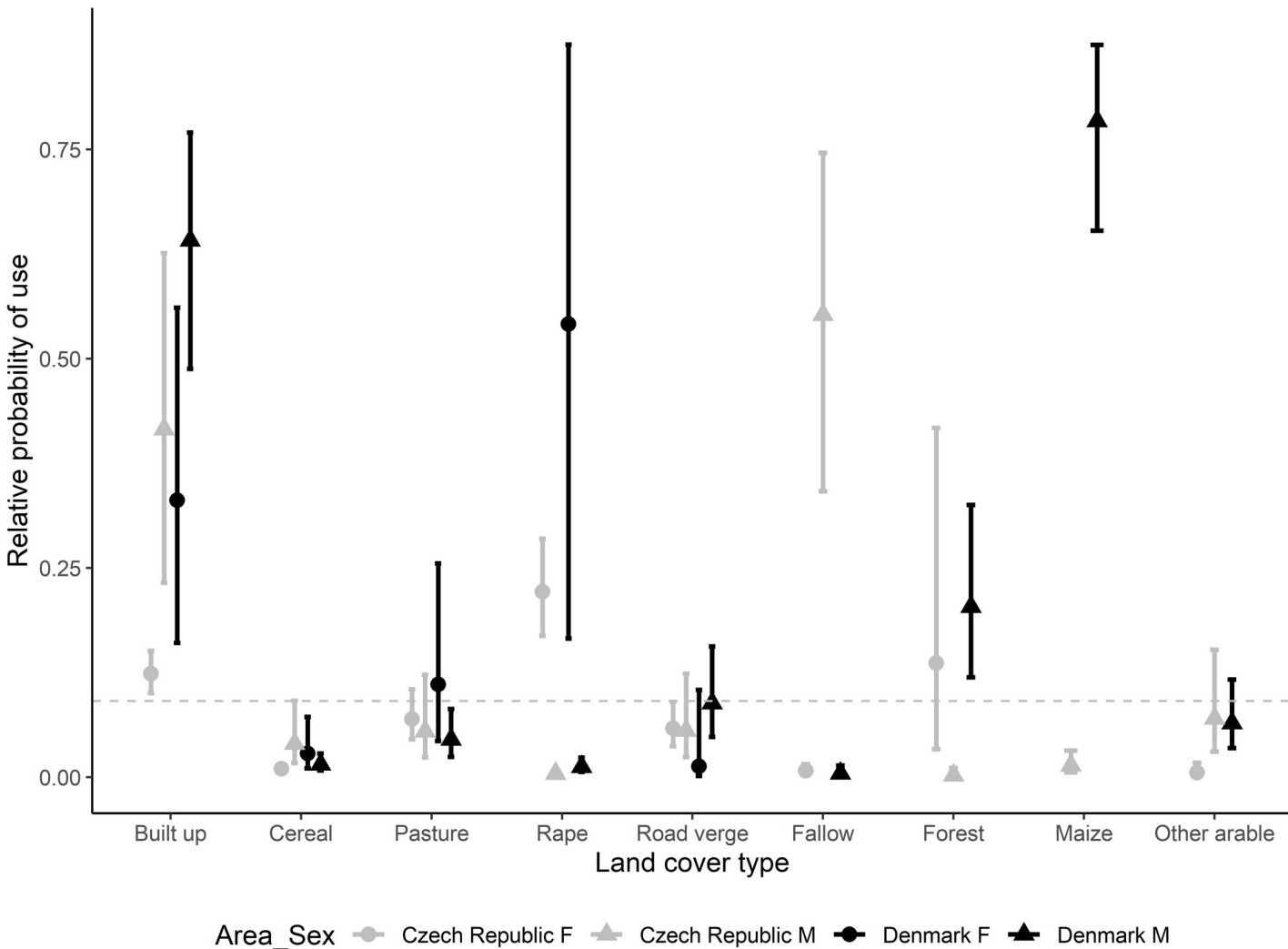

**Fig 5. The relative probability of use (as estimated by logistic resource selection functions) by Czech (grey) and Danish (black) little owls for the different land cover types.** Females are represented with circles and males with triangles. Available positions were created using correlated random walks. Values >0.1 (above the grey dashed line) indicate selection, whereas values <0.1 indicate avoidance (for each used position we created 10 random positions). The 95% confidence intervals are given as bars.

Denmark were much larger compared to Czech males and compared to previous studies [35, 60–63]. Increased foraging distances potentially impose greater energetic costs on the individual [64] and might affect adult survival due to excessive energy expenditure concentrated during the breeding season, as shown in other species [65]. The cause of the extended movements by Danish male owls likely reflects lower vertebrate prey availability, necessitating more energetic effort invested per prey item, i.e. longer search times. Consistent with this hypothesis, vertebrate prey in food-supplemented Danish owls only comprised ca. 34% of the natural prey biomass (as compared to ca. 47% earthworms and ca. 19% insects: [26]), suggesting that extended movements were driven by low vertebrate prey densities close to the nest. This is important, because decreased foraging efficiency impedes the efficiency of food provisioning to nestlings, ultimately affecting their survival [28, 29]. Large foraging distances by male owls also suggest that food provisioning alone was not sufficient to sustain the chick rearing, although it was previously shown to increase fledgling success [28]. In contrast, vole densities

were unusually high in the year of the study in the Czech Republic, allowing owls to meet their energetic needs in closer proximity to the nest. Owls in both areas moved greater distances when they had more chicks, indicating foraging effort increases with brood size, as found in other studies [66], although we should be prudent about concluding too much due to the low variation in the number of chicks (ranging from three to five) in this study. It is possible that the fledglings were not exactly the same age in the two areas (and among territories), which might have also affected owl movements due to varying food requirements of younger versus older fledglings. Weather also partly influenced owl movements, with Czech owls moving further from the nest when it was warmer and raining, potentially because they switched to alternative prey offered by different land cover types that were further from the nest [20]. Similarly, male owls in both areas moved further from the nest during earlier and later parts of the night (compared to the middle of their activity period), potentially to meet the energetic demands of the chicks (i.e. chicks may be more hungry in the early hours and may benefit of being fed up to satiation before daybreak) or due to variation in foraging opportunities (e.g. diel variation in prey activity and exposure) [27, 67]. We recognize that the study design (food supplementation in Denmark, but not in the Czech Republic) and the low sample of tagged birds did not allow to clearly disentangle the drivers of varying spatial movement patterns between the two countries, e.g. habitat composition, prey availability and food supplementation.

In general, little owls in both populations used foraging areas with either comparatively short vegetation (e.g. lawns, gardens and orchards in built up areas) that enabled hunting of ground-dwelling prey [35, 62] or areas with high structural diversity (close to field edges or fallow areas in the case of Czech males) that potentially support higher prey abundance [68, 69]. There were some differences between the two populations; e.g. males in the Czech Republic selected for fallow areas, and avoided maize and forest whereas the opposite pattern was found for Danish males. These differences might have been caused by fine-scale habitat variation (such as patch size, vegetation height and vegetation composition in the case of fallow areas) affecting foraging efficiency by owls. In particular, the proportion of ruderal vegetation around farmsteads was higher in the Czech Republic than Denmark, potentially providing better rodent habitat [70, 71]. One male in Denmark mostly foraged in a maize field with increasing distance from the nest, which consisted of low vegetation (<25 cm) at the time of the study, allowing that individual to hunt on the ground. In agricultural areas, proximity to roads and field edges likely facilitated foraging opportunities, as these habitats represent important habitats/refuges for small mammals as well as perches [68, 72]. Road surfaces also provide opportunities for owls to localize and capture large insects like carabid beetles in dry weather and earthworms and amphibians in wet weather [25] as well as invertebrate and small mammal road-kill carrion. Favorable foraging conditions close to roads may, however, increase collision risk with vehicles which is an important additive source of little owl mortality [73].

Our findings have implications for little owl conservation. The large movement distances of male owls in Denmark together with previous findings of rapid population declines [28, 29, 74] suggests that prey availability is insufficient to sustain the current Danish little owl population, even with the current regime of food supplementation. Owls in both areas generally selected for similar land cover types, i.e., areas with low open vegetation, and for proximity to field edges that likely provided perching opportunities and had an increased abundance of insects and rodents, especially those with hedges [68, 75, 76]. This implies that increasing habitat heterogeneity through the provision of short-sward grassland in association with high-quality edge habitats (e.g. hedgerows, fallows, field margins) to provide cover for prey [77] and perches for foraging birds will both increase prey availability and foraging success of little owls. Studies from other areas, where little owl home ranges were considerably smaller [34, 60, 62, 63], indicate that such habitat improvements can be highly effective at very small scales,

e.g. on the level of individual farms. In conclusion, this study demonstrates how high-resolution GPS data can be used to obtain information regarding fine-scale spatial movement patterns and habitat selection, which are important to inform management practices for conservation.

## Supporting information

**S1 Fig. Examples of simulated correlated random walks (grey lines) generated based upon real little owl trajectories (red lines).** The nest location is shown in each case as a yellow dot. We generated 10× the number of available data points (via random walks) than we had GPS positions (although here we only show the same number of used and available data for reasons of clarity and comparability).
(DOCX)

**S2 Fig.** The effect size of the interaction between the land cover type and distance from the nest on the relative probability of use, shown separately for male little owls from (a) Denmark and (b) the Czech Republic. Shading indicates 95% confidence intervals.
(DOCX)

**S1 Table. Overview of the best, full and intercept only model for the analyses of little owl distance from the nest and displacement per hour (for all data and separately for Danish and Czech owls).** $R^2m$ = marginal $R^2$ (fixed effects only), $R^2c$ = conditional $R^2$ (fixed effects and random intercept).
(DOCX)

**S2 Table. Effect size (β), standard error (SE), lower 95% confidence interval (LCI) and upper 95% confidence interval (UCI) of explanatory variables for the analyses of the (1) distance from nest and (2) displacement per hour.** Results are shown for the analyses of all data and separately for Danish (GPS-tagged in 2019) and Czech little owls (GPS-tagged in 2020). Informative values are given in bold (i.e., 95% confidence intervals not overlapping zero).
(DOCX)

**S3 Table. Effect size (β), standard error (SE), lower 95% confidence interval (LCI) and upper 95% confidence interval (UCI) of explanatory variables for the analyses of the (1) number of daily little owl foraging trips >200 m from the nest and (2) duration of foraging trips >200 m from the nest.** Informative values are given in bold (i.e., 95% confidence intervals not overlapping zero).
(DOCX)

## Acknowledgments

We thank Frank Skaarup Jensen and Johan Castenschiold (Denmark) and Miroslav Bažant, Lubomír Peške, Jiří Vlček, Petr Jandík, and Jitka Uhlíková (Czech Republic) for their help during the fieldwork. We are also grateful to all farmers and landowners for their hospitality and allowing us to work on their farmsteads during the research. Finally, we thank two reviewers for their constructive feedback that improved our manuscript.

## Author Contributions

**Conceptualization:** Anthony David Fox, Peter Sunde.

**Data curation:** Martin Šálek, Peter Sunde.

**Formal analysis:** Martin Mayer.

**Funding acquisition:** Martin Šálek, Peter Sunde.

**Investigation:** Martin Mayer, Martin Šálek, Anthony David Fox, Frej Juhl Lindhøj, Lars Bo Jacobsen.

**Methodology:** Martin Mayer, Martin Šálek, Anthony David Fox, Frej Juhl Lindhøj, Lars Bo Jacobsen, Peter Sunde.

**Project administration:** Peter Sunde.

**Resources:** Anthony David Fox, Peter Sunde.

**Visualization:** Martin Mayer.

**Writing – original draft:** Martin Mayer.

**Writing – review & editing:** Martin Šálek, Anthony David Fox, Lars Bo Jacobsen, Peter Sunde.

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
