## [Decision Letter · Decision Letter 0]

21 Jun 2021

PONE-D-21-11960

Fine-scale movement patterns and habitat selection of little owls (Athene noctua) from two declining populations

PLOS ONE

Dear Dr. Mayer,

Thank you for submitting your manuscript to PLOS ONE. After careful consideration, we feel that it has merit but does not fully meet PLOS ONE’s publication criteria as it currently stands. Therefore, we invite you to submit a revised version of the manuscript that addresses the points raised during the review process.

Your manuscript got positive review from both referees. I found that all their comments are useful to improve your work, then pay attention to all suggestions they gave to you. Especially to comments suggesting a better explanation about your small sample sizes and reasons for choosing the populations and individuals.

We look forward to receiving your revised manuscript.

Kind regards,

Paulo Corti, Ph.D.

Academic Editor

PLOS ONE

Journal Requirements:

3. We note that Figure 1 and Fig S1 in your submission contain map images which may be copyrighted. All PLOS content is published under the Creative Commons Attribution License (CC BY 4.0), which means that the manuscript, images, and Supporting Information files will be freely available online, and any third party is permitted to access, download, copy, distribute, and use these materials in any way, even commercially, with proper attribution. For these reasons, we cannot publish previously copyrighted maps or satellite images created using proprietary data, such as Google software (Google Maps, Street View, and Earth). For more information, see our copyright guidelines: http://journals.plos.org/plosone/s/licenses-and-copyright.

3.1.    You may seek permission from the original copyright holder of Figure 1 and Fig S1 to publish the content specifically under the CC BY 4.0 license. 

3.2.    If you are unable to obtain permission from the original copyright holder to publish these figures under the CC BY 4.0 license or if the copyright holder’s requirements are incompatible with the CC BY 4.0 license, please either i) remove the figure or ii) supply a replacement figure that complies with the CC BY 4.0 license. Please check copyright information on all replacement figures and update the figure caption with source information. If applicable, please specify in the figure caption text when a figure is similar but not identical to the original image and is therefore for illustrative purposes only.

Reviewers' comments:

Reviewer #1: Movement patterns of little owls - study discusses spatio temporal movement patterns of little owls using GPS in denmark and Czech Republic - supplemented and non-supplemented respectively.

Significant differences found between males & females in DK and males & m+f in CR, particularly in distance moved from nest, but reports more similar habitat selection between the two countries.

The authors used simulated correlated random walks to account for spatiotemporal autocorrelation.

Reviewer comments:

* Nice writing style, easy to read and well laid out.

* I would caution that although i completely understand that small sample sizes are the best that can be obtained sometimes, it's difficult to take a huge amount from 2 isolated populations. I think your reasons for choosing them could be better explained.

line 95-96 - would it also be useful to contrast with populations that are doing well?

128 - weight of little owls - so 3.2g is <2.5% but perhaps it is useful to mention if danish owls weight less than other little owls across Europe.

136 - would it not be better to have a scale of rainfall? yes/no doesn't then differentiate between light drizzle and a downpour.

could you include weight of the bird in your LMMs? then you can actually start to link energetic cost to distance etc?

286-288 - this result (interaction of area x sex) makes me think you should not have excluded the higher order

interaction from your other analysis (area x sex x time of night) - these interactions are interesting and perhaps there are differences due to chick brooding/ incubation?

lines 318 - 323 - this section is a bit list-like. Could you frame it a bit differently? the figure (5) does a great job of showing what you mean,

and i think it's a bit convoluted and hard to find out what's actually important. What i take away from it is that they select for different things,

in the 2 different countries and the sexes also differ.

line 330 -331 - this is a vague sentence, i think you could remove it. Start with the more interesting bit below that you have a clear indication that fixes of >15 mins apart would lose data.

(as you also reiterate in 336-338).

354-356 - I think this is also why trying to see if bodyweight had any influence would be useful. I note in your table you also provide number of chicks. Did this have any

influence on distance/time away from the nest?

line 361 - "..densities close to the nest."

373-374 - The fact that birds in DK are food supplemented does not actually feature very prominently throughout the methods and results section of the paper.

Perhaps it cannot, but by this point I had almost forgotten this is what you had done! You don't mention it when you talk about distance from the nest, etc.

What more can you say about its influence? So they still flew much further even though they were supplemented? I think there needs to be more information about this throughout.

378 - 380 - can you express structural diversity/crop height etc in one of your models using a metric of height differential or something like that? This is a great concluding

sentence - It would be good if we were able to understand this from the results. This adds to literature about little owls so perhaps you can use a DSM/DTM/Lidar to create a metric and

look at whether males/females in dk/cz select specifically for a strong height differential/short areas? Crop type as a factor does not always tell us this.

408 -409 - I don't think this is an impactful conclusion here. We KNOW that GPS can provide this information - this is well known, infact you state it in the intro.

Your impactful conclusions are those related to the habitat management and the stress that owls are under due to longer movements.

Potential other discussion comments: could you assess habitat available to populations in other areas? Do you know - since there are only 15 breeding pairs of little owls in DK, where they nest? And assess whether they are likely to behave in the same way?

Comments on figure legends - sometimes you put Czech birds first, sometimes Danish. I think you need consistency throughout, so either a is always Cz or Dk and b is always the other.

Reviewer #2: I enjoyed reading this manuscript that leverages very high resolution (one position per minute) GPS tracking data over a short period (~1 week) to better understand fine-scale habitat selection and nightly movement behavior of little owls in Denmark and the Czech Republic, two regions where populations are declining. They report on fine-scale habitat selection features that were consistent across the individuals studied in the two regions: areas with short vegetation and structural diversity that facilitate prey abundance and capture success. They then provide some conservation insights for this species, including suggesting that increasing habitat heterogeneity by providing both low grassland and high-quality edge (e.g. hedgerows) will improve foraging habitat for little owls. They also identify road verges as a potential ecological trap for this species.

Though the study has its limitations in terms of extrapolating findings to the broader populations due to limited sample size and sampling period (as acknowledged by the authors), this study provides important insight into the fine scale movements of a species that is declining in multiple regions across its range. The manuscript is well-written and interesting to read, the analysis appears to be soundly done (although see some of my comments for alternative approaches – perhaps for future publications). I have some suggestions below that I believe will improve the clarity of the manuscript for the broad readership of PLOS ONE.

Approach, aims and continuity: I felt that the aims of the project were a little hard to follow. From the title I was expecting a conservation focused paper, but the abstract seemed to indicate a methods paper. I would say the conservation approach is best, since you are not comparing the correlated random walk approach to other methods. Also, when we reached the predictions in the last paragraph of the introduction, it is not immediately clear to the reader how these specific predictions relate to the aims of the project. Similarly, it would be fantastic if it was simple for the reader to follow from your aims, to your predictions, through methods and results without having to check back on how it all fits together.

Methods: these comments are intended for the interest of the authors and not as a suggestion for re-analysis. I appreciated your use of correlated random walks in the habitat selection analysis and enjoyed reading about this method that I have not used before. However, autocorrelation in the movement data is not addressed in the other analyses in your study (e.g. distance travelled, home range etc.). In case you’re not familiar with the work, the continuous time movement model approach (Calabrese, Fleming, & Gurarie, 2016) is a method designed to account for the autocorrelation structures within movement data in order to help estimate these other movement metrics (Fleming et al., 2015; Noonan et al., 2019). In line with this, you may want to refer to your nightly KDE home ranges as “nightly occurrence” or “nightly space use” given these areas are dependent upon the sampling regime (Fleming et al., 2016) and they show a single night’s space use, rather than longer term ranging behavior. For example, they are likely to be much smaller when measured with a 1 min sampling interval compared to if they were measured with a one hour sampling interval. Like I mentioned above, I don’t see a need to re-analyze, but the authors may want to acknowledge the influence of autocorrelation with reference to their other metrics (where this is not accounted for in the analysis) in the Discussion and mention why it is unlikely to influence their interpretation of results.

Lines 39-41. Individual responses are not discussed in the paper.

Lines 39-41. What is “inferior foraging habitat”?

Line 98. Replace “excessively” (value judgement), could replace with “X-X times larger”, or “much larger”

Lines 94-104. Link these predictions to your aims, to understand space use in order to inform habitat restoration activities

Lines 124-129. Important to mention during exactly which part of the breeding season the owls were tracked (nest-building, incubation, fledging etc) since it was such a short period. We also need to know whether this was the same period in both regions.. It would be useful to comment more on this during the discussion too, e.g. consistency between life cycle periods sampled across regions and whether you’d expect to find something different during other parts of the year.

Line 222. Use “as” instead of “than”

Lines 289-291. This issue, the relationship between number of GPS fixes and displacement, could be addressed by taking the CTMM approach described above (Noonan et al., 2019).

Lines 344-376. Please make sure it’s clear whether you are talking about generalities in your results (same results across both study areas) or about results for one of the study areas. It may be useful to work this into your structure, e.g. speaking about generalities first and differences second (or vice versa).

Lines 358-361. Is this a diet analysis of food supplemented or un-supplemented owls?

Line 366. Add the range of number of chicks here

Line 387. Replace “providing” with “provided”.

Fig. 5. Could you please explain more about where the relative probability of use measurement comes from and how the avoidance threshold (grey dotted line) works?

Literature referred to:

Calabrese, J. M., Fleming, C. H., & Gurarie, E. (2016). Ctmm: an R Package for Analyzing Animal Relocation Data As a Continuous-Time Stochastic Process. Methods Ecol. Evol. 7, 1124–1132.

Fleming, C. H., Fagan, W. F., Mueller, T., Olson, K. A., Leimgruber, P., & Calabrese, J. M. (2016). Estimating where and how animals travel: an optimal framework for path reconstruction from autocorrelated tracking data. Ecology 97, 576–582.

Fleming, C. H., Fagan, W. F., Mueller, T., Olson, K. A., Leimgruber, P., Calabrese, J. M., & Cooch, E. G. (2015). Rigorous home range estimation with movement data: A new autocorrelated kernel density estimator. Ecology 96, 1182–1188.

Noonan, M. J., Fleming, C. H., Akre, T. S., Drescher-Lehman, J., Gurarie, E., Harrison, A. L., Kays, R., & Calabrese, J. M. (2019). Scale-insensitive estimation of speed and distance traveled from animal tracking data. Mov. Ecol. 7, 1–15.

3. Have the authors made all data underlying the findings in their manuscript fully available?

---

## [Author Response · Author response to Decision Letter 0]

7 Jul 2021

Dear Dr. Corti,

Thank you very much for giving us the opportunity to resubmit our manuscript. We would like to thank both reviewers for the time and effort they took to provide us with constructive feedback. We have incorporated their suggestions in the revised version. Below, we respond to each comment in detail.

We hope that these changes to the manuscript will facilitate the decision to publish our study in PLOS ONE.

Sincerely,

Martin Mayer

PONE-D-21-11960

Fine-scale movement patterns and habitat selection of little owls (Athene noctua) from two declining populations

PLOS ONE

Dear Dr. Mayer,

Thank you for submitting your manuscript to PLOS ONE. After careful consideration, we feel that it has merit but does not fully meet PLOS ONE’s publication criteria as it currently stands. Therefore, we invite you to submit a revised version of the manuscript that addresses the points raised during the review process.

Your manuscript got positive review from both referees. I found that all their comments are useful to improve your work, then pay attention to all suggestions they gave to you. Especially to comments suggesting a better explanation about your small sample sizes and reasons for choosing the populations and individuals.

We look forward to receiving your revised manuscript.

Kind regards,

Paulo Corti, Ph.D.

Academic Editor

PLOS ONE

Journal Requirements:

** We now updated the formatting of out manuscript accordingly.

** We have now uploaded relevant data to Dryad. The DOI is: https://doi.org/10.5061/dryad.k3j9kd57m

3. We note that Figure 1 and Fig S1 in your submission contain map images which may be copyrighted. All PLOS content is published under the Creative Commons Attribution License (CC BY 4.0), which means that the manuscript, images, and Supporting Information files will be freely available online, and any third party is permitted to access, download, copy, distribute, and use these materials in any way, even commercially, with proper attribution. For these reasons, we cannot publish previously copyrighted maps or satellite images created using proprietary data, such as Google software (Google Maps, Street View, and Earth). For more information, see our copyright guidelines: http://journals.plos.org/plosone/s/licenses-and-copyright.

** The images in Fig 1 and Fig S1 are not copyrighted. The inset map in Fig 1 comes from an open access data source and the land cover data was collected by us directly. We now added this in the figure caption of Fig 1. Fig S1 only includes data that was collected (or simulated by us).

** Corrected as suggested.

Reviewers' comments:

Reviewer #1

Movement patterns of little owls - study discusses spatio temporal movement patterns of little owls using GPS in denmark and Czech Republic - supplemented and non-supplemented respectively.

Significant differences found between males & females in DK and males & m+f in CR, particularly in distance moved from nest, but reports more similar habitat selection between the two countries.

The authors used simulated correlated random walks to account for spatiotemporal autocorrelation.

Reviewer comments:

* Nice writing style, easy to read and well laid out.

* I would caution that although i completely understand that small sample sizes are the best that can be obtained sometimes, it's difficult to take a huge amount from 2 isolated populations. I think your reasons for choosing them could be better explained.

** We thank the reviewer for the positive feedback, and have now expanded our reasoning for choosing the two different populations. Lines 92-97: ‘We compared nightly little owl home range size, spatio-temporal movement patterns, and habitat selection in surviving territories in Denmark with a local population in the Czech Republic that is also declining (Šálek and Schröpfer 2008), but not food-supplemented. We chose this comparison to contrast space use patterns among individuals of these two populations that both consisted of intensively cultivated farmland, but differed in the composition of crop types and regarding supplementary feeding, with the aim to understand space use to inform habitat restoration activities.’

line 95-96 - would it also be useful to contrast with populations that are doing well?

** Yes, we very much agree with the reviewer and actually planned to also GPS-tag individuals in an increasing population in Schleswig-Holstein, Germany. However, we unfortunately did not obtain the required ethics permits, and thus could not add this comparison.

128 - weight of little owls - so 3.2g is <2.5% but perhaps it is useful to mention if danish owls weight less than other little owls across Europe.

** We have no evidence for this statement. Our data regarding body mass from Danish little owls are not different from owls of other populations.

136 - would it not be better to have a scale of rainfall? yes/no doesn't then differentiate between light drizzle and a downpour.

** We considered this option. The problem is that there was no night with strong rainfall and only very few nights with some rain, so we opted for the categorization. Including precipitation as numeric variable would not have changed our results (we initially checked this).

could you include weight of the bird in your LMMs? then you can actually start to link energetic cost to distance etc?

**We agree with the reviewer that including body mass could be interesting to explain the variation in movement patterns between individuals (especially within same-sexed individuals). However, considering that we only have two to four observations per group (i.e. 3 M and 3 F in CZ, and 4 M and 2 F in DK), we doubt that the variation in body mass would have allowed us to make biologically relevant inferences. Additionally, the small sample size did not allow us to include this many variables in our analysis.

286-288 - this result (interaction of area x sex) makes me think you should not have excluded the higher order

interaction from your other analysis (area x sex x time of night) - these interactions are interesting and perhaps there are differences due to chick brooding/ incubation?

** We agree with the reviewer that these differences are interesting, but argue that we took these things into account by analyzing the data separately for the two study areas (making it easier for the reader to interpret the results as compared to three-way interactions).

lines 318 - 323 - this section is a bit list-like. Could you frame it a bit differently? the figure (5) does a great job of showing what you mean, and i think it's a bit convoluted and hard to find out what's actually important. What i take away from it is that they select for different things, in the 2 different countries and the sexes also differ.

** We have now slightly modified this part to make it more clear, and as suggested, we added a sentence to state the main message, i.e. that habitat selection differed between sexes and areas (lines 368-372): ‘Habitat selection partly differed between females and males, and between the two study areas. Relative to available locations obtained from correlated random walks (Fig. S1, 6), all owls selected for built up areas and avoided cereal, and neither selected for nor avoided pasture and road verges. Rape was avoided by males, but selected by females. Maize and forest were selected by Danish males and avoided by Czech males, and other arable crops were neither selected nor avoided by males.’

line 330 -331 - this is a vague sentence, i think you could remove it. Start with the more interesting bit below that you have a clear indication that fixes of >15 mins apart would lose data. (as you also reiterate in 336-338).

** We somewhat disagree with the reviewers comment on this point, and argue that this sentence gives a general summary of the relevance of our findings.

354-356 - I think this is also why trying to see if bodyweight had any influence would be useful. I note in your table you also provide number of chicks. Did this have any influence on distance/time away from the nest?

** The number of chicks did not affect how far from the nest owls moved. We now tested this by including the interaction of sex and body mass in the model investigating the distance from the nest, but this effect was uninformative (both the interaction and the main effect of body mass only). We would like to leave this variable out of the analyses to not overfit our models, considering the small sample size (which would not allow for a meaningful biological interpretation of the general effect of body mass). 

line 361 - "..densities close to the nest."

** Corrected as suggested.

373-374 - The fact that birds in DK are food supplemented does not actually feature very prominently throughout the methods and results section of the paper.

Perhaps it cannot, but by this point I had almost forgotten this is what you had done! You don't mention it when you talk about distance from the nest, etc.

What more can you say about its influence? So they still flew much further even though they were supplemented? I think there needs to be more information about this throughout.

** We now clarify in the methods that Czech owls were not food-supplemented (lines 118-119): ‘Czech little owls were not food-supplemented.’ And further in the results (lines 289-291): ‘In total, we obtained 14,639 GPS positions of 6 individuals in Denmark (food-supplemented; 4 males and 2 females) and 6 owls in the Czech Republic (not food-supplemented; 3 males and 3 females), with GPS units obtaining data for 1-8 nights (Table 1).’

378 - 380 - can you express structural diversity/crop height etc in one of your models using a metric of height differential or something like that? This is a great concluding sentence - It would be good if we were able to understand this from the results. This adds to literature about little owls so perhaps you can use a DSM/DTM/Lidar to create a metric and look at whether males/females in dk/cz select specifically for a strong height differential/short areas? Crop type as a factor does not always tell us this.

** This is a bit tricky, as we did not measure the exact vegetation height in all fields during the study. We doubt that such data would be obtainable via DTM or Lidar data (we are not aware of any such datasets at the relevant spatial scale) as the vegetation height is changing markedly from week to week during this time of the year. Thus, we are afraid that we have to restrict the discussion of vegetation height and structure based on vegetation type, although we generally agree with the reviewer that this would have been good to measure directly.

408 -409 - I don't think this is an impactful conclusion here. We KNOW that GPS can provide this information - this is well known, in fact you state it in the intro. Your impactful conclusions are those related to the habitat management and the stress that owls are under due to longer movements.

** With all due respect, we disagree with the reviewer on this point. The important message from this sentence is that fine-scale movement patterns obtained from GPS data can be used for conservation actions, which we elaborate in more detail in the preceding part of this paragraph. Thus, we would like to keep this sentence as is.

Potential other discussion comments: could you assess habitat available to populations in other areas? Do you know - since there are only 15 breeding pairs of little owls in DK, where they nest? And assess whether they are likely to behave in the same way?

** Considering that the habitat composition is similar in the other Danish areas where the remaining little owls nest, it is conceivable that they have similar space use patterns. However, without data to substantiate this speculation, we think that it makes little sense to include this in the discussion (also considering that the manuscript is quite long already; >8500 words).

Comments on figure legends - sometimes you put Czech birds first, sometimes Danish. I think you need consistency throughout, so either a is always Cz or Dk and b is always the other.

** We now changed the order in Fig. 2, so graphs for the Czech Republic consistently appear first in all figures.

Reviewer #2

I enjoyed reading this manuscript that leverages very high resolution (one position per minute) GPS tracking data over a short period (~1 week) to better understand fine-scale habitat selection and nightly movement behavior of little owls in Denmark and the Czech Republic, two regions where populations are declining. They report on fine-scale habitat selection features that were consistent across the individuals studied in the two regions: areas with short vegetation and structural diversity that facilitate prey abundance and capture success. They then provide some conservation insights for this species, including suggesting that increasing habitat heterogeneity by providing both low grassland and high-quality edge (e.g. hedgerows) will improve foraging habitat for little owls. They also identify road verges as a potential ecological trap for this species.

Though the study has its limitations in terms of extrapolating findings to the broader populations due to limited sample size and sampling period (as acknowledged by the authors), this study provides important insight into the fine scale movements of a species that is declining in multiple regions across its range. The manuscript is well-written and interesting to read, the analysis appears to be soundly done (although see some of my comments for alternative approaches – perhaps for future publications). I have some suggestions below that I believe will improve the clarity of the manuscript for the broad readership of PLOS ONE.

** We would like to thank the reviewer for the positive and constructive feedback.

Approach, aims and continuity: I felt that the aims of the project were a little hard to follow. From the title I was expecting a conservation focused paper, but the abstract seemed to indicate a methods paper. I would say the conservation approach is best, since you are not comparing the correlated random walk approach to other methods. Also, when we reached the predictions in the last paragraph of the introduction, it is not immediately clear to the reader how these specific predictions relate to the aims of the project. Similarly, it would be fantastic if it was simple for the reader to follow from your aims, to your predictions, through methods and results without having to check back on how it all fits together.

** We now partly revised our introduction to better link the background to our study aims and predictions. Lines 67-69: ‘Here, we conducted a case study on the little owl (Athene noctua) to investigate how high-frequency GPS data can be used to investigate spatio-temporal movement patterns and habitat selection to inform conservation practices.’

And further (lines 92-97): ‘We compared nightly little owl home range size, spatio-temporal movement patterns, and habitat selection in surviving territories in Denmark with a local population in the Czech Republic that is also declining (Šálek and Schröpfer 2008), but not food-supplemented. We chose this comparison to contrast space use patterns among individuals of these two populations that both consisted of intensively cultivated farmland, but differed in the composition of crop types and regarding supplementary feeding, with the aim to understand space use to inform habitat restoration activities.’

And finally (lines 105-107): ‘Regarding habitat selection, we predicted that owls generally select for areas that are associated with high prey densities and allow for successful foraging, i.e. comparatively short vegetation of pastures.’ 

Methods: these comments are intended for the interest of the authors and not as a suggestion for re-analysis. I appreciated your use of correlated random walks in the habitat selection analysis and enjoyed reading about this method that I have not used before. However, autocorrelation in the movement data is not addressed in the other analyses in your study (e.g. distance travelled, home range etc.). In case you’re not familiar with the work, the continuous time movement model approach (Calabrese, Fleming, & Gurarie, 2016) is a method designed to account for the autocorrelation structures within movement data in order to help estimate these other movement metrics (Fleming et al., 2015; Noonan et al., 2019). In line with this, you may want to refer to your nightly KDE home ranges as “nightly occurrence” or “nightly space use” given these areas are dependent upon the sampling regime (Fleming et al., 2016) and they show a single night’s space use, rather than longer term ranging behavior. For example, they are likely to be much smaller when measured with a 1 min sampling interval compared to if they were measured with a one hour sampling interval. Like I mentioned above, I don’t see a need to re-analyze, but the authors may want to acknowledge the influence of autocorrelation with reference to their other metrics (where this is not accounted for in the analysis) in the Discussion and mention why it is unlikely to influence their interpretation of results.

** We appreciate the reviewer’s suggestions regarding the continuous time movement model approach, which no doubt will be a useful tool for future movement analyses. Regarding the KDE definition, we already refer to them as ‘nightly home range size’, specifying that these estimates only refer to single nights. We agree of course that the KDE estimate could change depending on the sampling rate, but as we use the same fix rate across individuals and areas, this should not be an issue. Further, it was previously shown that KDE calculation is robust, given that the sample size is large enough (Börger et al. 2006, Wauters et al. 2007).

Lines 39-41. Individual responses are not discussed in the paper.

Lines 39-41. What is “inferior foraging habitat”?

** We have now removed these parts of the sentence.

Line 98. Replace “excessively” (value judgement), could replace with “X-X times larger”, or “much larger”

** We now write ‘at least twice the size’ to be more specific.

Lines 94-104. Link these predictions to your aims, to understand space use in order to inform habitat restoration activities

** We now revised this part to better link our study aims to the predictions. Lines 92-97: ‘We compared nightly little owl home range size, spatio-temporal movement patterns, and habitat selection in surviving territories in Denmark with a local population in the Czech Republic that is also declining (Šálek and Schröpfer 2008), but not food-supplemented. We chose this comparison to contrast space use patterns among individuals of these two populations that both consisted of intensively cultivated farmland, but differed in the composition of crop types and regarding supplementary feeding, with the aim to understand space use to inform habitat restoration activities.’

Lines 124-129. Important to mention during exactly which part of the breeding season the owls were tracked (nest-building, incubation, fledging etc) since it was such a short period. We also need to know whether this was the same period in both regions.. It would be useful to comment more on this during the discussion too, e.g. consistency between life cycle periods sampled across regions and whether you’d expect to find something different during other parts of the year.

** We now added the exact period (lines 136-137): ‘This period coincided with the fledgling period in both areas.’ However, we still did not know the exact age of the chicks, which might influence the food provisioning effort by the parents. As suggested by the reviewer, we now added this in the discussion (lines 424-426): ‘It is possible that the fledglings were not exactly the same age in the two areas (and among territories), which might have also affected owl movements due to varying food requirements of younger versus older fledglings.’

Line 222. Use “as” instead of “than”

** Corrected as suggested.

Lines 289-291. This issue, the relationship between number of GPS fixes and displacement, could be addressed by taking the CTMM approach described above (Noonan et al., 2019).

** We agree with the reviewer, but argue that our approach of including the number of GPS fixes into the model also accounts for biases in the estimation of hourly distances etc.

Lines 344-376. Please make sure it’s clear whether you are talking about generalities in your results (same results across both study areas) or about results for one of the study areas. It may be useful to work this into your structure, e.g. speaking about generalities first and differences second (or vice versa).

** We acknowledge that it was party unclear whether we were talking about a specific area or a general patterns, and now clarified these parts (please see within the manuscript).

Lines 358-361. Is this a diet analysis of food supplemented or un-supplemented owls?

** Food-supplemented owls. We added this information.

Line 366. Add the range of number of chicks here

** We added this.

Line 387. Replace “providing” with “provided”.

** We think the ‘providing’ is the correct term in this case.

Fig. 5. Could you please explain more about where the relative probability of use measurement comes from and how the avoidance threshold (grey dotted line) works?

** We have now added this information: ‘Available positions were created using correlated random walks. Values above the grey dashed line indicate selection, whereas values below the line indicate avoidance (for each used position we created 10 random positions).’

Literature referred to:

Calabrese, J. M., Fleming, C. H., & Gurarie, E. (2016). Ctmm: an R Package for Analyzing Animal Relocation Data As a Continuous-Time Stochastic Process. Methods Ecol. Evol. 7, 1124–1132.

Fleming, C. H., Fagan, W. F., Mueller, T., Olson, K. A., Leimgruber, P., & Calabrese, J. M. (2016). Estimating where and how animals travel: an optimal framework for path reconstruction from autocorrelated tracking data. Ecology 97, 576–582.

Fleming, C. H., Fagan, W. F., Mueller, T., Olson, K. A., Leimgruber, P., Calabrese, J. M., & Cooch, E. G. (2015). Rigorous home range estimation with movement data: A new autocorrelated kernel density estimator. Ecology 96, 1182–1188.

Noonan, M. J., Fleming, C. H., Akre, T. S., Drescher-Lehman, J., Gurarie, E., Harrison, A. L., Kays, R., & Calabrese, J. M. (2019). Scale-insensitive estimation of speed and distance traveled from animal tracking data. Mov. Ecol. 7, 1–15.

3. Have the authors made all data underlying the findings in their manuscript fully available?

References

Börger, L., N. Franconi, G. De Michele, A. Gantz, F. Meschi, A. Manica, S. Lovari, and T. Coulson. 2006. Effects of sampling regime on the mean and variance of home range size estimates. Journal of Animal Ecology 75:1393-1405.

Šálek, M., and L. Schröpfer. 2008. Recent decline of the Little Owl (Athene noctua) in the Czech Republic. Polish Journal of Ecology 56:527-534.

Wauters, L. A., D. G. Preatoni, A. Molinari, and G. Tosi. 2007. Radio-tracking squirrels: performance of home range density and linkage estimators with small range and sample size. Ecological modelling 202:333-344.

---

## [Decision Letter · Decision Letter 1]

2 Aug 2021

PONE-D-21-11960R1

Fine-scale movement patterns and habitat selection of little owls (Athene noctua) from two declining populations

PLOS ONE

Dear Dr. Mayer,

Thank you for submitting your manuscript to PLOS ONE. After careful consideration, we feel that it has merit but does not fully meet PLOS ONE’s publication criteria as it currently stands. Therefore, we invite you to submit a revised version of the manuscript that addresses the points raised during the review process.

Both reviewers provided useful comments, please provide arguments to their questions in some of their doubts, especially with reviewer #2.

We look forward to receiving your revised manuscript.

Kind regards,

Paulo Corti, Ph.D.

Academic Editor

PLOS ONE

Journal Requirements:

Reviewers' comments:

Review Comments to the Author

Reviewer #1: Thanks to the authors for addressing all of my comments on the first version of the manuscript.

Overall, i find your responses satisfactory and certainly understand your issue with small sample size affecting the addition of extra variables into your analyses such as weight of the birds.

I do think however, that the paper could benefit from explaining more about food supplementation. I do still feel that although you have made it clearer throughout about the supplementation, there is a lack of discussion about it. You state the findings of the paper have implications for conservation of little owls, which they definitely do, but perhaps state that even with supplementation they are struggling - so that is not a solution. So for e.g. lines 454 - 456 add on something like ", even with the current regime of supplementation" - because that's true isn't it? even with supplementation they are flying further and STILL not succeeding. I think perhaps the supplementation can be regarded as just an element of the study design but i think there is room to explore a bit why it's not working. You refer to the Kasper Thorup paper from 2010, early on where you state birds are food supplemented, but you could use this more to develop some discussion around it. I do realise you have stated your MS is long already, but i think you can do this succinctly and i think it would add something.

Further to this, I really only have a few minor comments on wording, and have indicated these below.

lines 96-97: your wording in this additional section you have added in is not quite right, where you have written ..'these two populations that both consisted of...' - the populations themselves do not consist of intensively cultivated famland, the areas in which they live do. So i would write something like "...these two populations, both of which are situated/ inhabit areas of intensively cultivated farmland, but which differ in exact compositon of crop types and in supplementation of feeding....'

lines 386 - 391: there is perhaps some literature that supports what you're saying here - see Mitchell et al. 2019 Plos One.

line 414: 'decreased' not 'decreasing'

Reviewer #2: The authors have done a good job of addressing most of my comments. My remaining suggestions are relatively minor and are mostly clarifications of my previous comments that need a little more consideration.

Regarding my comments on “nightly home range”, I still feel uncomfortable with the use of the language of “home range” here. The concept of a home range requires that an animal shows ranging behavior, e.g. crosses the same area multiple times and that the home range contains all resource requirements for that animal (Fleming et al., 2015; Calabrese, Fleming, & Gurarie, 2016). It is ultimately up to the Editor to make the decision here, but I think “nightly space use” would be more accurate and potentially less confounding to readers. Similarly, generally where distance travelled is reported in the literature as a straight-line distances between locations, it is referred to as “displacement” (Tucker et al., 2018; Noonan et al., 2019). These are changes suggested simply for clarity and consistency across the literature.

Regarding my comments on Fig. 5. To re-phrase more clearly, how was relative probability of use calculated? Please add to methods and briefly state here in the caption.

Regarding my previous comments and the authors’ response about generalities in results (pasted below):

“Lines 344-376. Please make sure it’s clear whether you are talking about generalities in your results (same results across both study areas) or about results for one of the study areas. It may be useful to work this into your structure, e.g. speaking about generalities first and differences second (or vice versa).

** We acknowledge that it was party unclear whether we were talking about a specific area or a general patterns, and now clarified these parts (please see within the manuscript).”

Please have another try at this. Firstly, it would be more convenient if you refer to whereabouts in the manuscript these changes have been made, as for other responses. From looking through the results, I can find only one sentence that has been changed:

“Habitat selection partly differed between females and males, and between the two study areas. Relative to available locations obtained from correlated random walks (Fig 6 and S1 Fig), all owls selected for built up areas and avoided cereal, and neither selected for nor avoided pastures and road verges.”

Unfortunately, this makes the distinction between study areas and sexes even less clear. If habitat selection differed between females and males as well as the two study areas, what does “neither” refer to?

Please address my above comments by first reading through and identifying the multiple locations across the results where this confusion occurs (unclear whether results apply to one study area or both, or one sex or both), address them clearly and indicate in the response where in the manuscript this has been done.

Lines 259-260. “We fitted generalized linear mixed-effects models with a logit link…” I assume with the binomial family, add this.

Calabrese, J. M., Fleming, C. H., & Gurarie, E. (2016). Ctmm: an R Package for Analyzing Animal Relocation Data As a Continuous-Time Stochastic Process. Methods Ecol. Evol. 7, 1124–1132.

Fleming, C. H., Fagan, W. F., Mueller, T., Olson, K. A., Leimgruber, P., Calabrese, J. M., & Cooch, E. G. (2015). Rigorous home range estimation with movement data: A new autocorrelated kernel density estimator. Ecology 96, 1182–1188.

Noonan, M. J., Fleming, C. H., Akre, T. S., Drescher-Lehman, J., Gurarie, E., Harrison, A. L., Kays, R., & Calabrese, J. M. (2019). Scale-insensitive estimation of speed and distance traveled from animal tracking data. Mov. Ecol. 7, 1–15.

Tucker, M. A., Böhning-Gaese, K., Fagan, W. F., Fryxell, J. M., Van Moorter, B., Alberts, S. C., Ali, A. H., Allen, A. M., Attias, N., Avgar, T., Bartlam-Brooks, H., Bayarbaatar, B., Belant, J. L., Bertassoni, A., Beyer, D., Bidner, L., Van Beest, F. M., Blake, S., Blaum, N., Bracis, C., Brown, D., De Bruyn, P. J. N., Cagnacci, F., Calabrese, J. M., Camilo-Alves, C., Chamaillé-Jammes, S., Chiaradia, A., Davidson, S. C., Dennis, T., DeStefano, S., Diefenbach, D., Douglas-Hamilton, I., Fennessy, J., Fichtel, C., Fiedler, W., Fischer, C., Fischhoff, I., Fleming, C. H., Ford, A. T., Fritz, S. A., Gehr, B., Goheen, J. R., Gurarie, E., Hebblewhite, M., Heurich, M., Hewison, A. J. M., Hof, C., Hurme, E., Isbell, L. A., Janssen, R., Jeltsch, F., Kaczensky, P., Kane, A., Kappeler, P. M., Kauffman, M., Kays, R., Kimuyu, D., Koch, F., Kranstauber, B., LaPoint, S., Leimgruber, P., Linnell, J. D. C., López-López, P., Markham, A. C., Mattisson, J., Medici, E. P., Mellone, U., Merrill, E., De MirandaMourão, G., Morato, R. G., Morellet, N., Morrison, T. A., Díaz-Muñoz, S. L., Mysterud, A., Nandintsetseg, D., Nathan, R., Niamir, A., Odden, J., O’Hara, R. B., Oliveira-Santos, L. G. R., Olson, K. A., Patterson, B. D., De Paula, R. C., Pedrotti, L., Reineking, B., Rimmler, M., Rogers, T. L., Rolandsen, C. M., Rosenberry, C. S., Rubenstein, D. I., Safi, K., Saïd, S., Sapir, N., Sawyer, H., Schmidt, N. M., Selva, N., Sergiel, A., … Mueller, T. (2018). Moving in the Anthropocene: Global reductions in terrestrial mammalian movements. Science (80-. ). 359, 466–469.

---

## [Author Response · Author response to Decision Letter 1]

3 Aug 2021

Reviewers' comments:

Review Comments to the Author

Reviewer #1: Thanks to the authors for addressing all of my comments on the first version of the manuscript.

Overall, i find your responses satisfactory and certainly understand your issue with small sample size affecting the addition of extra variables into your analyses such as weight of the birds.

** We would like to thank the reviewer for this positive feedback.

I do think however, that the paper could benefit from explaining more about food supplementation. I do still feel that although you have made it clearer throughout about the supplementation, there is a lack of discussion about it. You state the findings of the paper have implications for conservation of little owls, which they definitely do, but perhaps state that even with supplementation they are struggling - so that is not a solution. So for e.g. lines 454 - 456 add on something like ", even with the current regime of supplementation" - because that's true isn't it? even with supplementation they are flying further and STILL not succeeding. I think perhaps the supplementation can be regarded as just an element of the study design but i think there is room to explore a bit why it's not working. You refer to the Kasper Thorup paper from 2010, early on where you state birds are food supplemented, but you could use this more to develop some discussion around it. I do realise you have stated your MS is long already, but i think you can do this succinctly and i think it would add something.

** We agree with this comment, and now added some more considerations in the discussion regarding food supplementation. Lines 432-433: ‘Large foraging distances by male owls also suggest that food provisioning alone was not sufficient to sustain the chick rearing, although it was previously shown to increase fledgling success [28].’ And further (lines 471-474): ‘The large movement distances of male owls in Denmark together with previous findings of rapid population declines [28, 29, 72] suggests that prey availability is insufficient to sustain the current Danish little owl population, even with the current regime of food supplementation.’

Further to this, I really only have a few minor comments on wording, and have indicated these below.

lines 96-97: your wording in this additional section you have added in is not quite right, where you have written ..'these two populations that both consisted of...' - the populations themselves do not consist of intensively cultivated farmland, the areas in which they live do. So i would write something like "...these two populations, both of which are situated/ inhabit areas of intensively cultivated farmland, but which differ in exact composition of crop types and in supplementation of feeding....'

** Corrected as suggested.

lines 386 - 391: there is perhaps some literature that supports what you're saying here - see Mitchell et al. 2019 Plos One.

** Yes, we have now added this reference.

line 414: 'decreased' not 'decreasing'

** Corrected as suggested.

Reviewer #2: The authors have done a good job of addressing most of my comments. My remaining suggestions are relatively minor and are mostly clarifications of my previous comments that need a little more consideration.

** We would like to thank the reviewer for this positive feedback and additional suggestions for clarification.

Regarding my comments on “nightly home range”, I still feel uncomfortable with the use of the language of “home range” here. The concept of a home range requires that an animal shows ranging behavior, e.g. crosses the same area multiple times and that the home range contains all resource requirements for that animal (Fleming et al., 2015; Calabrese, Fleming, & Gurarie, 2016). It is ultimately up to the Editor to make the decision here, but I think “nightly space use” would be more accurate and potentially less confounding to readers. Similarly, generally where distance travelled is reported in the literature as a straight-line distances between locations, it is referred to as “displacement” (Tucker et al., 2018; Noonan et al., 2019). These are changes suggested simply for clarity and consistency across the literature.

** We agree and have now adjusted the wording (space use instead of home range size and displacement instead of distance moved) throughout the manuscript.

Regarding my comments on Fig. 5. To re-phrase more clearly, how was relative probability of use calculated? Please add to methods and briefly state here in the caption.

** The relative probability of use was calculated based on estimates obtained from logistic resource selection functions. We have now added this in the methods and give a reference that gives a detailed background (lines 262-264): ‘We fitted logistic resource selection functions, using generalized linear mixed-effects models with a binomial distribution and a logit link, using the R package ‘lme4’ [50]’. And in the figure legend: ‘The relative probability of use (as estimated by logistic resource selection functions) by Czech (grey) and Danish (black) little owls for the different land cover types.’

Regarding my previous comments and the authors’ response about generalities in results (pasted below):

“Lines 344-376. Please make sure it’s clear whether you are talking about generalities in your results (same results across both study areas) or about results for one of the study areas. It may be useful to work this into your structure, e.g. speaking about generalities first and differences second (or vice versa).

** We acknowledge that it was party unclear whether we were talking about a specific area or a general patterns, and now clarified these parts (please see within the manuscript).”

Please have another try at this. Firstly, it would be more convenient if you refer to whereabouts in the manuscript these changes have been made, as for other responses. From looking through the results, I can find only one sentence that has been changed:

“Habitat selection partly differed between females and males, and between the two study areas. Relative to available locations obtained from correlated random walks (Fig 6 and S1 Fig), all owls selected for built up areas and avoided cereal, and neither selected for nor avoided pastures and road verges.”

Unfortunately, this makes the distinction between study areas and sexes even less clear. If habitat selection differed between females and males as well as the two study areas, what does “neither” refer to?

Please address my above comments by first reading through and identifying the multiple locations across the results where this confusion occurs (unclear whether results apply to one study area or both, or one sex or both), address them clearly and indicate in the response where in the manuscript this has been done.

** We agree that this was still unclear and have now reworked the entire paragraph to clarify which owls we refer to in which sentence. Lines 373-389: ‘Regarding habitat selection (comparing used positions to available positions obtained from correlated random walks; Fig 6 and S1 Fig), all owls (of both sexes and study areas) strongly selected for built up areas, avoided cereal, and showed no selection or avoidance for pastures and road verges. For the other land cover types, habitat selection partly differed between females and males, and between the two study areas. In both areas, rape was avoided by males, but selected by females. Maize and forest were selected by Danish males and avoided by Czech males, and other arable crops were neither selected nor avoided by males of both populations. Fallow, maize, and other arable crops were either avoided by females (in both populations) or not available; i.e., not present in proximity to the nest (Fig 5). Forest was neither selected for nor avoided by Danish females and not available to Czech females. Danish males selected for proximity to roads whereas Danish females avoided proximity to roads (this variable was not included in the analyses of Czech owls), and all owls, apart from Czech females, selected for proximity to field edges (Table 2). Finally, male owls in Denmark increasingly selected for maize with increasing distance from the nest at the expense of the other land cover categories, whereas there was no clear pattern for males in the Czech Republic (S2 Fig).’

Lines 259-260. “We fitted generalized linear mixed-effects models with a logit link…” I assume with the binomial family, add this.

** Corrected as suggested.

---

## [Decision Letter · Decision Letter 2]

11 Aug 2021

Fine-scale movement patterns and habitat selection of little owls (Athene noctua) from two declining populations

PONE-D-21-11960R2

Dear Dr. Mayer,

We’re pleased to inform you that your manuscript has been judged scientifically suitable for publication and will be formally accepted for publication once it meets all outstanding technical requirements.

Kind regards,

Paulo Corti, Ph.D.

Academic Editor

PLOS ONE

**Comments to the Author**

1. If the authors have adequately addressed your comments raised in a previous round of review and you feel that this manuscript is now acceptable for publication, you may indicate that here to bypass the “Comments to the Author” section, enter your conflict of interest statement in the “Confidential to Editor” section, and submit your "Accept" recommendation.

Reviewer #2: All comments have been addressed

2. Is the manuscript technically sound, and do the data support the conclusions?

Reviewer #2: Yes

3. Has the statistical analysis been performed appropriately and rigorously? 

Reviewer #2: Yes

4. Have the authors made all data underlying the findings in their manuscript fully available?

Reviewer #2: Yes

5. Is the manuscript presented in an intelligible fashion and written in standard English?

Reviewer #2: Yes

---

## [Editor Report · Acceptance letter]

9 Sep 2021

PONE-D-21-11960R2 

Fine-scale movement patterns and habitat selection of little owls (*Athene noctua*) from two declining populations 

Dear Dr. Mayer:

I'm pleased to inform you that your manuscript has been deemed suitable for publication in PLOS ONE. Congratulations! Your manuscript is now with our production department. 

Kind regards, 

on behalf of

Dr. Paulo Corti 

Academic Editor

PLOS ONE